# The Diverse Potential of Gluten from Different Durum Wheat Varieties in Triggering Celiac Disease: A Multilevel In Vitro, Ex Vivo and In Vivo Approach

**DOI:** 10.3390/nu12113566

**Published:** 2020-11-20

**Authors:** Federica Gaiani, Sara Graziano, Fatma Boukid, Barbara Prandi, Lorena Bottarelli, Amelia Barilli, Arnaldo Dossena, Nelson Marmiroli, Mariolina Gullì, Gian Luigi de’Angelis, Stefano Sforza

**Affiliations:** 1Gastroenterology and Endoscopy Unit, University Hospital of Parma, University of Parma, via Gramsci 14, 43126 Parma, Italy; federica.gaiani@unipr.it; 2Interdepartmental Center Biopharmanet-tec, Parco Area delle Scienze, University of Parma, 43124 Parma, Italy; lorena.bottarelli@unipr.it; 3Interdepartmental Center SITEIA.PARMA, Parco Area delle Scienze, University of Parma, 43124 Parma, Italy; sara.graziano@unipr.it (S.G.); fatma.boukid@unipr.it (F.B.); barbara.prandi@unipr.it (B.P.); arnaldo.dossena@unipr.it (A.D.); nelson.marmiroli@unipr.it (N.M.); 4Department of Food and Drug, Parco Area delle Scienze, University of Parma, 27/A-43124 Parma, Italy; 5Department of Medicine and Surgery, Unit of Pathological Anatomy, University Hospital of Parma, via Gramsci 14, 43126 Parma, Italy; 6Department of Medicine and Surgery, Unit of General Pathology, University of Parma, Via Volturno 39, 43125 Parma, Italy; amelia.barilli@unipr.it; 7Department of Chemistry, Life Sciences, and Environmental Sustainability, University of Parma, Parco Area delle Scienze 11a, 43124 Parma, Italy

**Keywords:** celiac disease, durum wheat, gluten peptides, immune response, ELISA

## Abstract

The reasons behind the increasing prevalence of celiac disease (CD) worldwide are still not fully understood. This study adopted a multilevel approach (in vitro, ex vivo, in vivo) to assess the potential of gluten from different wheat varieties in triggering CD. Peptides triggering CD were identified and quantified in mixtures generated from simulated gastrointestinal digestion of wheat varieties (*n* = 82). Multivariate statistics enabled the discrimination of varieties generating low impact on CD (e.g., Saragolla) and high impact (e.g., Cappelli). Enrolled subjects (*n* = 46) were: 19 healthy subjects included in the control group; 27 celiac patients enrolled for the in vivo phase. Celiacs were divided into a gluten-free diet group (CD-GFD), and a GFD with Saragolla-based pasta group (CD-Sar). The diet was followed for 3 months. Data were compared between CD-Sar and CD-GFD before and after the experimental diet, demonstrating a limited ability of Saragolla to trigger immunity, although not comparable to a GFD. Ex vivo studies showed that Saragolla and Cappelli activated immune responses, although with great variability among patients. The diverse potential of durum wheat varieties in triggering CD immune response was demonstrated. Saragolla is not indicated for celiacs, yet it has a limited potential to trigger adverse immune response.

## 1. Introduction

Celiac disease (CD) is a chronic autoimmune enteropathy triggered by dietary gluten in genetically predisposed individuals [1]. In Western countries, the prevalence of CD in the general population is about 1%, with regional differences, such as in Finland and Sweden, where prevalence rises to 2–3% [2]. Epidemiologic data have shown a constant increase in incidence of CD not only in countries where a high number of gluten-containing products are the basis of dietary habits, but also in Asian countries such as India [3]. The exact reasons driving this increasing incidence are not yet clear: nowadays, diagnostic techniques are easily available all over the world, and both general practitioners and pediatricians are more and more experienced in suspecting and screening the disease, but certainly global changes in dietary habits related to higher consumption of wheat-based food are playing a role in modifying CD epidemiology [4]. 

CD development depends on the presence of key genes that regulate the immunological response to dietary gluten and, in particular in subjects bearing human leucocyte antigen (HLA) DQ2/DQ8 haplotypes, gluten represents the key environmental factor triggering the immune response that is the basis of small-intestine mucosal damage, mediated both by innate and adaptive immunity [5,6]. The presence of these alleles is necessary but not sufficient for disease development: although the HLA-DQ2 allele is common in the white population (30% of people are carriers), almost 3% of them will develop CD [6]. On the contrary, 95% of CD patients carry the HLA-DQ2 (DQA1*0501/DQB1*0201) haplotype, while the remaining patients express the HLA-DQ8 (DQA1*0301/DQB1*0302) haplotype [7].

Gluten is a group of glutamine and proline-rich proteins called prolamins, contained in the grains of several cereals, including wheat, rye, barley and oats. In wheat, these proteins are called gliadins or glutenins, based on their monomeric or polymeric structure, and together they form gluten [8]. After gluten ingestion, proteases contained in gastrointestinal secretions hydrolyze glutamine- and proline-rich gluten-composing proteins [9]. However, gluten peptides are not completely hydrolyzed by gastrointestinal enzymes because of their high amounts of proline. Adjuvated by increased permeability of the intestinal mucosa [10], partially hydrolyzed proteins reach the lamina propria, where they are modified by tissue transglutaminase (tTG) by deamidation and transamidation, becoming more affine to major histocompatibility complex class II (MHCII) and initiating lymphocyte activation [9]. Genetic predisposition is necessary to develop CD, indeed HLA-DQ2 and/or HLA-DQ8 alleles code for specific MHC molecules, which are surface receptors on antigen-presenting cells belonging to the immune system. The HLA-DQ2 and HLA-DQ8 proteins have high affinity for deamidated, negatively charged gluten-derived peptides [11]. Although it is not clear how and when the autoimmune reaction driven by gluten is initiated, a high amount of gluten-derived peptides and a high frequency in dietary gluten consumption increases the probability of having substrates to initiate the inflammatory process at the basis of CD.

Several approaches have been followed to better understand CD. In vitro gastrointestinal digestion models have been widely used, with different systems, such as single static, multicompartmental and dynamic. Quite obviously, diverse approaches generated different results, making the comparison between studies not possible [12,13,14]. A standard model was recently developed to simulate digestive processes [15]. Quantitative tools were afterward required for the quantification of total gluten or the generated peptides associated with CD, such as enzyme-linked immunosorbent assays and proteomics tools [16,17]. The peptides obtained from gliadin digestion trigger the inflammation of the intestinal mucosa due to both the adaptive and the innate immune responses. Intestinal biopsy cultures from celiac patients were demonstrated to be useful in determining innate and adaptive responses to gluten [18]. In particular, interleukin (IL)-15 and interferon (IFN)-γ are part of a more complex network which is responsible of the dysregulation of multiple immune mechanisms in the small intestine, together contributing to CD pathogenesis [19]. It is noteworthy that, even if in vitro and ex vivo approaches have provided great understanding of CD, in vivo studies remain crucial to fully understand how the ingestion of gluten impacts the gastrointestinal tract. 

From a multilevel perspective, the present study aims to explore: (i) at the in vitro level, the high diversity in durum wheat varieties on the amount of peptides associated with CD, generated after simulated gastrointestinal digestion, in order to identify a variety with less triggering peptide production, thus lowering the impact on the immune response; (ii) at the ex vivo level, the potential of the above gluten-derived peptides to trigger the production of inflammatory cytokines usually implicated in the initiation of autoimmune inflammation leading to CD and (iii) at the in vivo level, the comparison of a gluten-free diet with a gluten-free diet with a controlled amount of food produced with a selected wheat species, identified as above for being low impact, measuring the immune activation with tTG IgA serum antibodies. The research of a durum wheat variety with low impact on the immune response does not aim to implement a gluten-free diet for celiac subjects, who are not allowed to take gluten, rather it is addressed to the general non-celiac population’s dietary habits. The use of wheat varieties that are less stimulating to the immune system, compared to wheat varieties traditionally employed in food production, could bring potential benefits for patients predisposed to celiac disease and not yet manifesting the disorder.

## 2. Materials and Methods

### 2.1. In Vitro Study

#### 2.1.1. Material 

The durum wheat varieties (*Triticum durum* Desf.; *n* = 82 varieties) utilized in this research were cultivated in the field at the Council for Agricultural Research and Economics–Research Centre for Cereal and Industrial Crops (CREA-CI), Foggia (Italy, 41° 28′ N, 15° 32′ E and 75 m a.s.l.) in a clay–loam soil (typic chromoxerert) in the growing season of 2015−2016. Seeds after harvesting were milled using a laboratory mill (Ika Werke, Staufen, Germany) and stored in plastic bags at 4 °C until analysis. Pasta made with Saragolla wheat and used in this study had a water content of 0.092 ± 0.001%, and a protein content of 10.8 ± 0.3% (wet basis). Immunogenic peptides after in vitro digestion were 1363 ± 66 ppm, and toxic peptides were 461 ± 27 ppm.

#### 2.1.2. In Vitro Gastrointestinal Digestion

Whole wheat flours were subjected to simulated gastrointestinal digestion [15] as follows: (i) 1 g of flour was incubated (2 min; 37 °C under constant gentle mixing) with 1 mL simulated saliva containing porcine amylase (Sigma-Aldrich, St. Louis, MI, USA; 75 U/mL of digesta); (ii) 2 mL simulated gastric juice containing porcine pepsin (Sigma-Aldrich, St. Louis, MI, USA; 2000 U/mL of digesta) was added and incubated (2 h; 37 °C under constant gentle mixing) after pH adjustment to 3; (iii) 4 mL duodenal juice containing porcine pancreatin (SigmaAldrich, St. Louis, MI, USA; 100 U trypsin activity/mL of digesta) and porcine bile (Sigma-Aldrich, St. Louis, MI, USA; 10 mmol/L in the total volume) was added and incubated (2 h; 37 °C under constant gentle mixing) after pH adjustment to 7. Afterward, to inactivate the enzymes, the sample was boiled for 10 min at 95 °C. After centrifugation (3220 g, 4 °C, 45 min), the supernatant (295 μL) was added to 5 μL of internal standard solution (TQQPQQPF(d5)PQQPQQPF(d5)PQ; 1.6 mM); samples were then subjected to reverse phase liquid chromatography coupled to mass spectrometry (RP-UPLC/ESI-MS) to quantify peptides associated with CD [20]. All the samples were digested in duplicate.

### 2.2. In Vivo Study

In the period between November 2017 and November 2018, all the patients who admitted at the Gastroenterology and Endoscopy Unit of the University Hospital of Parma, Italy, to undergo esophagogastroduodenoscopy for suspected CD were evaluated for inclusion in the present study. The inclusion criteria were age between 2 and 75 years, elevated tTG IgA (UNL (upper limits of normal) = 10 U/mL) and or DPG-AGA IgG (UNL = 10 U/mL). A cohort of patients comparable for demographics and admitted at the Gastroenterology and Endoscopy Unit of the University Hospital of Parma, Italy, for reasons other than suspected CD was considered for inclusion in the control group (CG). These patients were evaluated for eligibility while being referred to the hospital to undergo esophagogastroduodenoscopy for gastroesophageal reflux symptoms, dyspepsia, abdominal pain or weight loss. Their inclusion in the CG was confirmed once duodenal biopsies taken during endoscopy were found to be normal, therefore excluding celiac disease.

Eligible patients and their families were given information describing the study and, subsequently, written consent to participate was gathered. The study protocol was conducted in accordance with the Declaration of Helsinki (1964) and the protocol was approved by the Ethics Committee of Parma (Project POR-FESR 2014–2020) on 25 July 2017. As an ethical requirement, patients were included only if able to sign informed consent (by the legal guardian in case of minors).

Each patient was characterized by demographics and clinical data, including serum levels of hemoglobin, total IgA levels, tTG IgA, DGP-AGA IgG, HLA haplotype for suspected celiac patients, ongoing treatments, first-degree familiarity for CD and comorbidities; all the included patients underwent duodenal biopsies to ascertain CD in suspected patients and to rule it out in controls. Specifically, all controls underwent one biopsy in the descending duodenum and one biopsy in the bulb, while for suspected celiac patients four biopsies in the descending duodenum and one in the bulb were taken, in accordance with current guidelines for diagnosis [21]. Moreover, all the included patients underwent three additional biopsies taken from the descending duodenum for the in vitro study.

After diagnosis of CD was ascertained by histologic examination and graded in accordance with Marsh–Oberhüber classification [22], celiac patients were divided in two cohorts on a voluntary basis. The first cohort followed a gluten-free diet including a daily portion (40–70 g of pasta based on the age of the considered patient) of experimental pasta made with Saragolla wheat flour (CD-Sar cohort) for three months; pasta was provided by the investigators. The second cohort followed a complete and rigorous gluten-free diet (CD-GFD cohort). Both the CD-Sar and the CD-GFD cohorts underwent serum dosage of the same parameters described at the enrollment after three months of the diet. Experimental pasta samples used were produced at the pilot plant of Barilla (Parma, Italy) following the conventional industrial processing diagram of pasta production. None of the patients either in the CD-GFD group nor in the CD-Sar group had ever followed a gluten-free diet before inclusion in the study. Nutritional counseling was provided by gastroenterologists after diagnosis of CD and before the start of the study, to maximize compliance to the diet (either totally gluten-free or gluten-free with a portion of Saragolla wheat pasta). 

Data extracted from demographics, antibody serum levels before and after three months of the diet, HLA typing and histological examination were compared among groups.

### 2.3. Ex Vivo Study

All the included patients underwent three additional biopsies taken from the descending duodenum for the ex vivo study. Freshly isolated biopsies were placed in the wells of a tissue culture plate containing DMEM/F12 supplemented with 10% heat-inactivated fetal calf serum, 100 U/mL penicillin and 100 mg/mL streptomycin. From each patient, one mucosa biopsy was cultured with the medium (control), one was cultured in the medium after the addition of 5% of the in vitro digested flour of Cappelli and one was cultured after the addition of 5% of the in vitro digested flour of Saragolla. The culture plate was placed in an organ culture chamber, gassed with a mixture of 95% O_2_ and 5% CO2, and incubated at 37 °C for 4 h. As demonstrated by previous works [23], significant differences in the cytokine production were already measurable after 120 min of biopsy incubation with digested gliadin. So, the incubation period was set to 4 h to maximize the possibility to observe eventual differences in the cytokine production among the samples. At the end of the culture period, supernatant fluid from the cultured specimens was collected and stored at −70 °C until used. IL-15 and IFN-γ were determined in the supernatants by ELISA using the Quantikine^®^ ELISA Human IL-15 Immunoassay and Human IFN-γ Immunoassay (R&D Systems, Inc., Minneapolis, MN 55413, USA), following the manufacturer’s instructions.

### 2.4. Statistical Analysis 

Significant differences among the generated peptides after digestion were found using analysis of variance (ANOVA) with a confidence interval of 95% (*p* ≤ 0.05). Principal component analysis based on unsupervised features (peptides generated after in vitro digestion) was performed based on the correlation matrix. For the evaluation of significant differences of the output of in vivo and ex vivo results, the non-parametric Wilcoxon–Mann–Whitney test was used (*p* ≤ 0.05). All the statistical analyses were determined using the program SPSS for Windows (Version 24.0, SPSS Inc., Chicago, IL, USA).

## 3. Results

### 3.1. In Vitro Results 

The peptides identified could be subdivided into two groups, according to the literature [24,25,26]: peptides triggering the adaptive and innate immune responses. Seven immunogenic peptides derived from γ-gliadin were identified as being involved in the adaptive immune response (“immunogenic peptides”, IPs). IP sequences are described in Figure 1. All IP sequences contained the motif DQ2.5-glia-γ4c (QQPQQPFPQ) [27]. Three peptides deriving from α-gliadin were identified as being involved in the innate immune response (“toxic peptides”, TPs). TP sequences are described in Figure 1. The sequence of these peptides contained the motifs PSQQ, QQQP, QQPY or QPYP able to trigger the innate immune response. The quantification of these peptides showed significant differences, as illustrated in Figure 1 and based on statistical data (Appendix A). Full characterization of the 82 varieties can be found in Taranto et al. (2020) [28]. Such a high variability might be attributed to high genetic variability among the studied varieties (i.e., matrix properties including protein structure and starch−protein interactions), which greatly influence digestive enzyme accessibility to gluten [29,30].

Principal component analysis (PCA) was therefore performed to enable a better picture. Figure 2A displays the two-dimensional scattering plot of the quantity of peptides associated with CD. The first two principal components (PCs) explained 83% of the total variability. The first component explained 60% as a function of the major part of the identified peptides (IP1, IP2, IP3, IP4, IP5, IP6, IP7, TP1 and TP2); whereas the second component explained only 23% as a function of TP3 (Figure 2A). The projection of the studied durum wheat samples on the factorial space (Figure 2B) showed important variability among the 82 varieties, confirming the high genetic diversity of the studied collection. 

Using PCA (Figure 2B), we were also able to locate varieties with a low amount of triggering peptides after digestion, thus with a potentially low impact on CD (Group 1) and those with a high amount of triggering peptides after digestion, thus with high impact (Group 2). As shown in Figure 2B, two groups of wheat varieties were clearly identified: high in CD-triggering peptides (total peptides between 900 ppm and 2700 ppm) and low in CD-triggering peptides (total peptides <600 ppm). The selection of Saragolla and Cappelli was based on the amount of peptides, Saragolla (513 ppm peptides generated after digestion) was selected as representative of Group 1, and Cappelli (1488 ppm peptides generated after digestion) as representative of Group 2, to be used in the following analyses. Both varieties are widely cultivated in Italy and utilized to produce monovarietal flours for pasta; Saragolla is a modern variety, released in 2004, while Cappelli is an old one, released in 1915. 

### 3.2. In Vivo Results 

Overall, 46 patients with ages between 3 and 69 years were recruited: 19 non-celiac patients were included in the control group (CG), while 27 celiac patients were divided into the CD-Sar group (11 patients) and CD-GFD group (16 patients). All the patients included in the CD-Sar group were minors (aged between 3 and 17 years), while among patients included in CD-GFD group, 13 were minors and three were adults (aged 45, 20 and 25 years). Characteristics of the included patients are described in Table 1.

Most of the recruited patients were females (60.87%), who represented most patients in the CD-GFD group, while sexes were equally distributed among the CD-Sar and CG. In all groups, most of the patients had no comorbidities and were on no medications. The distribution of symptoms of presentation was inhomogeneous and only a few celiac patients presented with typical CD (e.g., diarrhea, weight loss). At recruitment, tTG IgA levels were elevated and suspicious for CD, with comparable titers between the CD-Sar group and CD-GFD group. None of the patients diagnosed with celiac disease presented selective IgA deficiency. First-degree familiarity for celiac CD was negative in most celiac patients, and in all cases in the control group. Overall, HLA haplotype distribution was inhomogeneous and did not show significant differences (*p* = 0.085) between groups. Regarding diagnosis of CD, histology of duodenal biopsies was investigated in all groups, showing a complete normality in the CG, and the highest degree of pathognomonic histologic characteristics for CD both in the CD-Sar and CD-GFD groups, with duodenal biopsies classified as Marsh 3 grade in 81.82% and 93.75% of patients, respectively. For all celiac patients, no other histologic lesions were detected in gastric and esophageal biopsies, except for one patient, who was demonstrated to be affected by eosinophilic esophagitis. This patient was included in the CD-Sar group. Overall, the CD-Sar and CD-GFD groups demonstrated comparable clinic characteristics.

The effect of the inclusion in the diet of pasta prepared with Saragolla wheat flour on immune activation was then compared to the GFD, by the determination of antibody serum levels before and after 3 months of the diet. Overall, two patients included in the CD-Sar cohort and eight patients included in the CD-GFD cohort refused to undergo the second antibody determination after 3 months of the diet. Results are shown in Table 2. 

Specifically, tTG IgA variation (Δ) after the administered diet was determined and compared between the CD-Sar and CD-GFD groups, showing no significant differences (p=0.06). Results are shown in Table 2. 

The influence of diet on immune activation was mainly determined by tTG IgA variation, although DPG-AGA IgG and hemoglobin serum levels were also investigated, showing similarly no differences between groups (Table 2).

In the patient diagnosed as celiac and affected by eosinophilic esophagitis, it was also possible to compare duodenal biopsies before and after three months of the diet with Saragolla wheat, as the patient needed to repeat endoscopy after therapy with proton pump inhibitors, accordingly to international guidelines [31]. As shown in Figure 3 and Figure 4, diet with Saragolla wheat pasta did not modify duodenal histology, and it did not worsen the Marsh grade, which was 3a both at diagnosis and after three months of the diet. The comparison between the gluten-free diet with Saragolla wheat and the completely gluten-free diet was carried out for a limited period of time (3 months) with the only aim to observe if the immune response of celiac patients continued to be triggered by Saragolla wheat as by a traditional wheat-containing diet, or if this selected wheat variety had a limited potential to trigger autoimmunity, in terms of tTG titer. Yet, we specify that the cohort of celiac patients undergoing the gluten-free diet with Saragolla was necessary, as celiac patients are the only subjects in which an immune response is measurable by using antibody titers (tTG IgA). We also specify that the diet including Saragolla was stopped after three months for celiac patients, as it is not suitable for them to achieve remission of the disease.

### 3.3. Ex Vivo Results 

Overall, 43 patients were subjected to three additional duodenal biopsies which were utilized for the ex vivo treatments with the products of the in vitro digestion of rice flour (GF) and Saragolla (Sar) and Cappelli (Cap) wheat flours. In many studies, organ culture experiments have been performed by using digested gliadin fractions because many more components are present in digested flour that could interfere with the results of the organ culture system [32,33]. However, using a digested flour is much closer to what physiologically happens in the human body than the selected peptide fraction. The biopsies utilized were from 16 non-celiac individuals (CG group) and 27 celiac patients (CD group). After the different treatments, the level of IL-15 and IFN-γ were measured by an ELISA test. In general, high individual variability was observed, for both IL15 and IFN- γ levels. Therefore, the values of IL-15 and of IFN-γ measured in biopsies treated with Sar or Cap were normalized with respect to GFD (Table 3). 

The effect of the Sar and Cap treatments was not significantly different in CG for both IL-15 and IFN-γ levels. However, Cap treatment determined an increase in IL-15 (20%) and IFN-γ (29%) levels, differently from the Sar treatment. Regarding the CD group, in 15 cases, the treatment with either Sar or Cap digestion products determined an increase in the amount of IL-15 of 48 and 43%, respectively, as compared with the GF treatment. In 12 cases, the IL-15 levels were similar in response to all the treatments with the in vitro digestion products. As for the IFN-γ levels measured in the CD group, Sar treatment determined an increase of about 87% with respect to GF treatment. For one celiac patient affected by eosinophilic esophagitis, it was also possible to compare duodenal biopsies before and after three months of the diet with Saragolla pasta. The results obtained showed that IL-15 levels remained stable in response to each treatment even after the diet; interestingly, the IFN-γ level increased after the treatment with Cap, but not with Sar, in biopsies taken after three months of the diet (Table 4).

## 4. Discussion

In recent decades, CD has changed from a rare disease to a more and more diagnosed disorder through the worldwide population, not only in Western countries, but also among Eastern populations such as India, where gluten consumption is limited, although increasing [3]. Even if the disease is well known in terms of pathognomonic histologic lesions and clinical manifestations, and the diagnostic approach is standardized all over the world, little is known about the mechanisms of initiation of the autoimmune response. Gluten is recognized as the main trigger of inflammation, and, at present, a completely gluten-free diet is the only effective treatment for celiac patients, and potentially the only way to avoid the initiation of the immune response. Nevertheless, based on dietary habits, especially in Western countries, a rigorous gluten-free lifelong diet is not always easy to follow. On the other hand, no effective preventive method of the onset of CD has been discovered. 

Starting from this background, the interest of this study was to investigate if selected wheat varieties had a different impact on the onset of the immune response leading to manifest CD. 

The pre-clinical in vitro phase was crucial to select the wheat variety, which produced the lowest number of immunogenic peptides. An important collection of durum wheat varieties was in vitro digested and analyzed by liquid chromatography coupled to mass spectrometry for their peptides triggering immune and innate responses. The identification of CD-triggering peptides after simulated gastrointestinal digestion and untargeted MS analysis revealed that the identified peptides belong mainly to γ-gliadin and α-gliadin. These findings are consistent with previous observations [20,34]. A high variability, determined by one-way analysis of variance (ANOVA), and further confirmed by multivariate analysis (PCA), was evident in the analyzed samples, suggesting that different varieties might bear strong differences in their potential to trigger CD. Two groups were identified, classified as “high” and “low” in CD-triggering peptides. Results revealed that Saragolla (Group 1, low impact) and Cappelli (Group 2, high impact) are the median of groups 1 and 2, respectively. Furthermore, Saragolla was also used for the next in vivo experiments. Although a challenge with pasta exclusively made with Cappelli variety would have been advisable, ethical concerns prevented this possibility.

Obviously, Saragolla wheat, as for all gluten-containing wheat species, cannot be integrated in a gluten-free diet, as demonstrated by the comparison between duodenal histology before and after the experimental diet (Figure 3 and Figure 4), as the typical lesions of CD are neither healed nor improved by this type of diet. The choice to administrate the experimental diet to celiac patients was led by the need for testing immunogenicity in a population which was certainly reactive to gluten peptides, and therefore celiac, but also sensitive to a reduction of exposure to the inflammatory trigger in terms of a reduction of specific antibody production. This method allowed us to avoid a gluten challenge in a cohort of celiac patients already on a gluten-free diet, and therefore in remission, as it would have raised ethical concerns. We started from a high and certainly pathological titer, considering that the absence of immune stimulus leads to a decrease in antibodies, and we evaluated how much the Saragolla diet detaches itself from the stimulation of the usual mix of grains and how much it can be assimilated into a gluten-free diet. Since tTG titer is not a reliable indicator of the severity of histologic lesions but an epiphenomenon of the autoimmune process at the basis of CD depending on the ability of each CD patient’s immune system to elevate antibodies [35], the evaluation of the tTG-IgA level improvement was not evaluated as absolute numbers, but as percent variation from the starting level (Δ).

The effects of Saragolla diet on histology could be measured only for one patient requiring a second endoscopy and, therefore, these results are isolated and do not allow us to draw strong conclusions. Nevertheless, our study demonstrated that Saragolla wheat did not worsen previous histologic lesions, which was quite surprising, but perfectly in line with our in vitro findings, demonstrating that Saragolla wheat produces, upon digestion, a low amount of CD-triggering peptides. As a controlled amount of gluten was administered through the Saragolla pasta, we did not expect the histological lesions of this patient to heal, but this patient remained clinically stable during the three months of experimentation, at Marsh 3a grade, and did not worsen to Marsh 3c grade. Moreover, the tTG IgA value decreased from 15 U/mL to 8.6 U/mL. Although a second duodenal biopsy would have been methodologically important to complete the clinical consequences of the Saragolla diet for all the patients included in the CD-Sar cohort, it would have needed a second invasive exam against any clinical indications, which would have been ethically inappropriate. Therefore, the comparison between tTG IgA values before and after the experimental diet was considered appropriate. Importantly, the gluten-free diet integrated with Saragolla wheat pasta did not worsen the immune response, as shown by the improvement or steadiness of antibody titers, which, albeit not pathogenic, are strongly linked to an ongoing autoimmune process. Overall, during the study, tTG IgA values demonstrated high variability both at the time of recruitment and after the three months of the diet, especially in the CD-Sar cohort, which highlight that antibody titers should never be considered as an absolute value, but always considered by means of variation of the value in time, probably due to a variability among patients in gluten tolerance and the ability of the immune system to produce specific autoantibodies.

The time period of three months for the experimentation is certainly limited, but it was chosen as an appropriate compromise between the need to observe variation of the immune response by means of antibody titer variation and ethical concerns regarding the deliberate intake of gluten among patients diagnosed as celiac (CD-Sar cohort) without exposing patients to the risk of deteriorating clinical conditions.

Of note, the study was not conducted as blind nor as double-blind, and therefore lacks a cohort undergoing placebo. Every patient included in the CD-Sar group needed to add in the diet a portion of pasta provided by the researchers, therefore it would have been hard to blind this specific action. Among the weak points of the study, celiac patients enrolled were not randomly assigned to either the CD-GFD cohort or CD-Sar cohort. This choice was made to guarantee the best possible compliance of the patients to the diet, who were willing to actively participate in either cohort. The impossibility to blind the inclusion in groups was a weak point of the study, although the effect of either a completely gluten-free diet or the “Saragolla diet” was measured rigorously by dosing tTG IgA titers before and after three months of the diet in both groups. Moreover, nutritional counseling was provided by gastroenterologists after the diagnosis of celiac disease and before the start of the diet, in order to maximize the compliance to the diet, therefore allowing the results of antibody titer dosage to show the real evolution of the immune response under the diet.

Based on the above results, it might be possible to deduce that the immune response is significantly less stimulated by Saragolla wheat, compared to what is known in the literature when common wheat varieties or gliadin extracts are used [33,36]. Actually, if we consider the natural history of CD, a celiac patient who is not following a strict gluten-free diet will continue having rises in tTG IgA values until the environmental trigger is eliminated, which is gluten, as an epiphenomenon of the perpetuation of the immune response trigger. Therefore, the trend shown by tTG IgA in the CD-Sar cohort highlighted a limited ability of Saragolla wheat to activate autoimmunity. The variation in antibody titer was unfortunately affected by a percentage of patients who dropped out during the follow-up, as shown in Table 2. Anyway, the obtained results allowed us to compare cohorts reliably, from a methodological perspective.

Even though Saragolla wheat administration is not indicated for celiac patients, it could be extremely interesting to hypothesize its employment in potentially at-risk individuals, such as first-degree relatives of celiac patients or those patients affected by autoimmune diseases related to CD (Down syndrome, Hashimoto thyroiditis, diabetes mellitus type 1, etc.). 

Certainly, the hypothesis to prevent CD onset by introducing less immune-reactive durum wheat varieties in the general population’s diet has not been proven yet. Anyway, the present study shows for the first time in the literature an experimental basis for this hypothesis, which will have to be confirmed in larger cohorts to be considered appliable to the general population, but is not contradicted by our results. Moreover, no previous comprehensive study has ever been reported on this topic, because it is very challenging to design and conduct clinical trials using food produced with monovarietal flours, which is one of the strong methodological points of our study.

The hypothesis of using Saragolla wheat, or varieties with similar characteristics, for the production of monovarietal flours and products with low immunogenicity might be considered. The possibility to spread, in large-scale commercial distribution, this kind of product, could, theoretically, reduce the burden of the best-known environmental trigger of CD, with a consequent lower incidence of the disease itself. 

This concept can be compared to the use of hydrolyzed milk in neonates, which has a reduced antigenic burden compared to conventional milk; it can be employed in the diet of unweaned babies with a familiarity for atopy, reducing the risk of cow’s milk protein allergy, a fearsome consequence of the early contact between the neonate’s intestine and heterologous proteins [35,37]. Although it is a theoretical assumption, the most evident benefit of the use of a less immunogenic wheat in the diet should be noted in Western countries, whose dietary habits are characterized by an important quantity of gluten-containing food.

## 5. Conclusions

The results of the present study demonstrate a limited ability of selected wheat varieties such as Saragolla to trigger the immune response, as the basis of CD in vitro, in vivo and ex vivo. The aim of this work was not to make gluten-free pasta but to demonstrate a minor impact of certain wheat varieties. However, the final target of a diet based on selected wheat varieties are not celiac patients, but those subjects at risk of CD. Although Saragolla cannot be applied in the gluten-free diet, in a future perspective, its employment could be hypothesized in large-scale commercial distribution, with the aim to reduce the burden of the environmental trigger of CD, therefore potentially reducing its incidence in the general population. This hypothesis has not been proven yet, but the present study provides for the first time in the literature an experimental basis to support it, which anyway will have to be confirmed in larger cohorts to better explore the feasibility of this solution. However, these first results are encouraging, demonstrating that this approach deserves further investigation.

## Figures and Tables

**Figure 1 nutrients-12-03566-f001:**
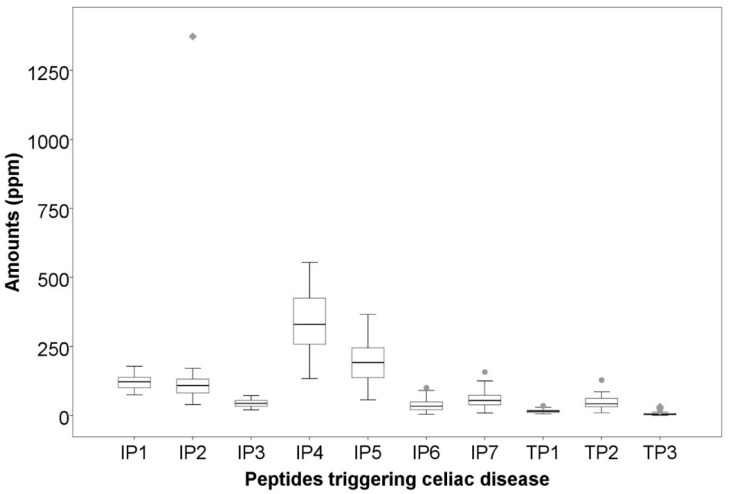
Variability in the amounts of peptides associated with celiac disease. IP1: TQQPQQPFPQ; IP2: SQQPQQPFPQPQ; IP3: QAFPQQPQQPFPQ; IP4: TQQPQQPFPQQPQQPFPQ; IP5: PQTQQPQQPFPQFQQPQQPFPQPQQP; IP6: FPQQPQLPFPQQPQQPFPQPQQPQ; IP7: QQPQQPFPQPQQTFPQQPQLPFPQQPQQPF. TP1: LQPQNPSQQQPQ; TP2: RPQQPYPQPQPQ. TP3: LQPQNPSQQQPQEQVPL.

**Figure 2 nutrients-12-03566-f002:**
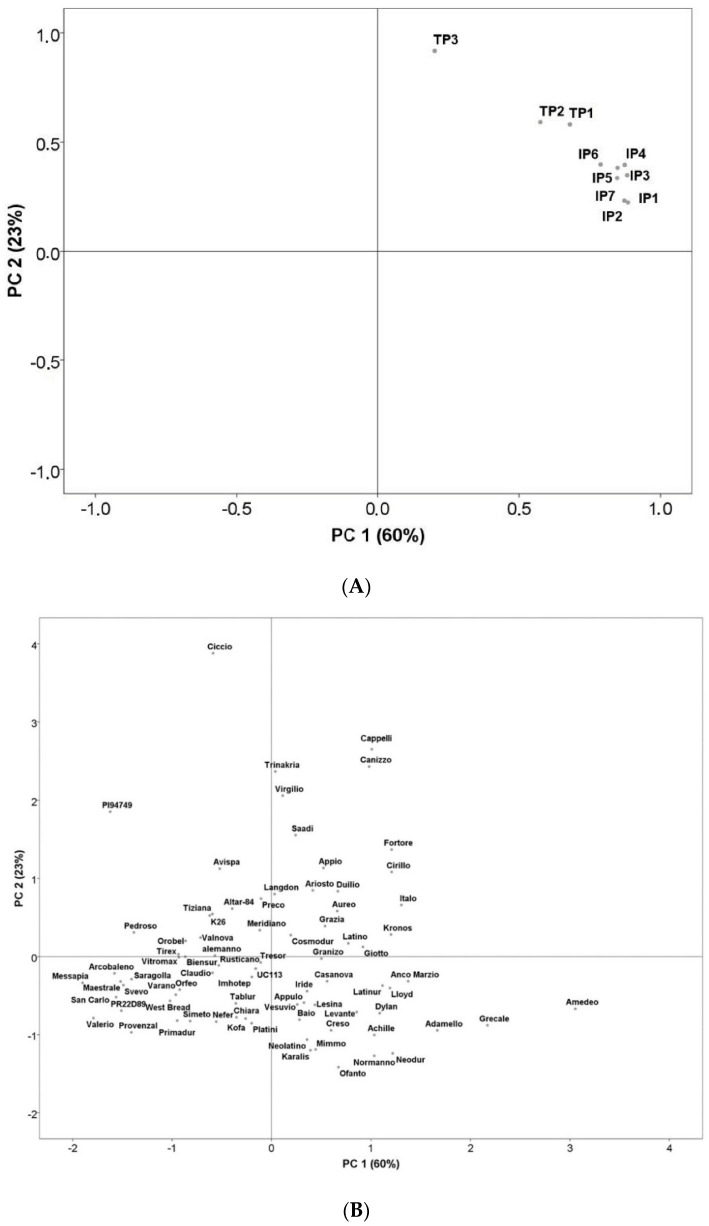
(**A**) Principal component analysis of durum wheat epitopes involved in celiac disease (CD). Biplot of the two first principal components based on the durum wheat epitopes involved in CD. Abbreviations IP1: TQQPQQPFPQ; IP2: SQQPQQPFPQPQ; IP3: QAFPQQPQQPFPQ; IP4: TQQPQQPFPQQPQQPFPQ; IP5: PQTQQPQQPFPQFQQPQQPFPQPQQP; IP6: FPQQPQLPFPQQPQQPFPQPQQPQ; IP7: QQPQQPFPQPQQTFPQQPQLPFPQQPQQPF. TP1: LQPQNPSQQQPQ; TP2: RPQQPYPQPQPQ. TP3: LQPQNPSQQQPQEQVPL. (**B**) Principal component analysis of durum wheat epitopes involved in CD. Rotated principal scores of the durum wheat varieties projected into the first two principal components.

**Figure 3 nutrients-12-03566-f003:**
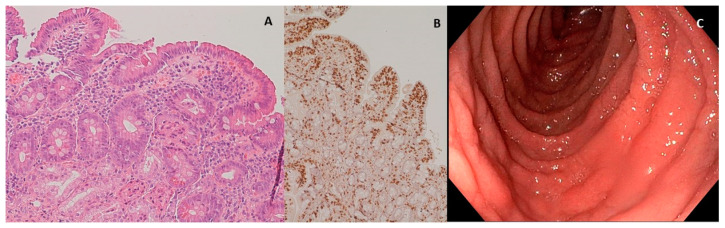
Duodenal mucosa before experimental diet. (**A**) Hematoxylin–eosin 4× magnification, duodenal mucosa before experimental diet. Celiac disease classified as Marsh 3a; (**B**) immunohistochemical coloration for CD3+ lymphocytes, duodenal mucosa before experimental diet; (**C**) endoscopic appearance of the duodenal mucosa before experimental diet.

**Figure 4 nutrients-12-03566-f004:**
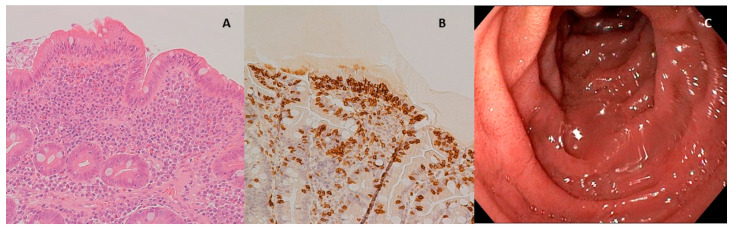
Duodenal mucosa after experimental diet. (**A**) Hematoxylin–eosin 4× magnification, duodenal mucosa after experimental diet. Celiac disease classified as Marsh 3a; (**B**) immunohistochemical coloration for CD3+ lymphocytes, duodenal mucosa after experimental diet; (**C**) endoscopic appearance of the duodenal mucosa after experimental diet.

**Table 1 nutrients-12-03566-t001:** Clinical characteristics of the included subjects.

Parameter	Total (*n* = 46)	CD-Sar (*n* = 11)	CD-GFD (*n* = 16)	CG (*n* = 19)
Age, years (mean (range))	(3–69)	8.27 (3–17)	12.96 (5–45)	29.42 (5–69)
**Sex**: M	18 (39.13%)	5 (45.45%)	3 (18.75%)	10 (52.63%)
F	28 (60.87%)	6 (54.55%)	13 (81.25%)	9 (47.37%)
**BMI** (mean (range))	(11.16–29.35)	16.41 (12.5–23.18)	16.4 (12–21.5)	19.58 (11.16–29.35)
**Comorbidities**:				
none	43 (93.49%)	11 (100%)	14 (87.5%)	18 (94.74%)
atopy	1 (2.17%)	0	1 (6.25%)	0
autoimmune thyroiditis	1 (2.17%)	0	1 (6.25%)	0
hematologic disorders	1 (2.17%)	0	0	1 (5.26%)
**Ongoing treatments**:				
none	40 (86.96%)	11 (100%)	13 (81.25%)	16 (84.21%)
PPI/ranitidine	3 (6.52%)	0	1 (6.25%)	2 (10.53%)
prokinetics	1 (2.17%)	0	0	1 (5.26%)
levothyroxine	1 (2.17%)	0	1 (6.25%)	0
oral contraceptives	1 (2.17%)	0	1 (6.25%)	0
**Symptoms**:				
none	9 (19.56%)	2 (18.18%)	7 (43.75%)	0
diarrhea	6 (13.04%)	1 (9.09%)	4 (25%)	1 (5.26%)
abdominal pain	14 (30.43%)	5 (45.45%)	6 (37.5%)	3 (15.79%)
growth delay/weight loss	7 (15.22%)	2 (18.18%)	3 (18.75%)	2 (10.53%)
constipation	3 (6.52%)	2 (18.18%)	1 (6.25%)	0
reflux symptoms/pyrosis	9 (19.56%)	0	0	9 (47.37%)
nausea/dyspepsia	4 (8.7%)	0	0	4 (21.05%)
**Familiarity for CD**:				
yes	7 (15.22%)	4 (36.36%)	3 (18.75%)	0
no	39 (84.78%)	7 (63.64%)	13 (81.25%)	19 (100%)
**tTG IgA** (mean (range))	n/a	79.68 (13–173)	85.11 (9.8–275)	n/a
**DPG-AGA IgG** (mean (range))	n/a	77.6 (3.3–302)	38.5 (3.3–82)	n/a
**Hb** (g/dL) (mean (range))	n/a	12.17 (7.4–14)	12.9 (11.5–15)	n/a
**HLA haplotype**:				
DQB1*02 ho/DQ8-	n/a	3 (27.3%)	1 (6.25%)	n/a
DQB1*02 he/DQ8+		1 (9%)	0	
DQB1*02 he/DQ8-		2 (18.2%)	6 (37.5%)	
DQB1*02 -/DQ8+		3 (27.3%)	0	
n/a		2 (18.2%)	9 (56.25%)	
**Marsh** (histology):				
0	19 (41.31%)	0	0	19 (100%)
1	3 (6.52%)	2 (18.18%)	1 (6.25%)	0
3 (a or b or c)	24 (52.17%)	9 (81.82%)	15 (93.75%)	0

Abbreviations: CD-Sar, group of celiac patients undergoing a gluten-free diet with pasta produced with Saragolla wheat; CD-GFD, group of celiac patients undergoing a gluten-free diet; CG, control group; M, males; F, females; BMI, body mass index; PPI, proton pump inhibitor; tTG IgA, antibody anti-transglutaminase, class IgA; DPG-AGA IgA, antibody anti-deaminated peptide of gliadin, class IgG; HLA, Human Leukocyte Antigen; Hb, hemoglobin; ho, homozygous; he, heterozygous. Familiarity stands for first-degree familiarity.

**Table 2 nutrients-12-03566-t002:** Variation of serum CD antibodies and hemoglobin values before and after 3 months of diet. Comparison between CD-Sar and CD-GFD.

Laboratory Test	CD-Sar (*n* = 11)	CD-GFD (*n* = 16)	Δ Sar-GFD
**tTG IgA**: T0 (mean (range))	79.68 (13–173)	85.11 (9.8–275)	
3 M (mean (range))	51.32 (7.1–114)	10.6 (2.8–24)	*p* = 0.06
n/a T0	0	0	
n/a 3 M	2 (18.2%)	8 (50%)	
**DPG-AGA IgG**: T0 (mean (range))	77.6 (3.3–302)	38.5 (3.3–82)	
3 M (mean (range))	20.8 (1.5–128)	6.65 (1.5–21)	*p* = 0.62
n/a T0	4 (26.4%)	8 (50%)	
n/a 3 M	3 (27.3%)	10 (52.6%)	
**Hb**: T0 (mean (range))	12.17 (7.4–14)	12.9 (11.5–15)	
3 M (mean (range))	12.28 (9–14.4)	13.5 (11.4–15.2)	*p* = 0.97
n/a T0 (%)	1 (9.1%)	3 (%)	
n/a 3 M (%)	3 (27.3%)	8 (50%)	

Abbreviations: CD-Sar, group of celiac patients undergoing a gluten-free diet with pasta produced with Saragolla wheat; CD-GFD, group of celiac patients undergoing a gluten-free diet; tTG IgA, antibody anti-transglutaminase, class IgA; DPG-AGA IgA, antibody anti-deaminated peptide of gliadin, class IgG; Hb, hemoglobin; T0, at the start of the study, time 0; 3 M, after three months of experimentation; n/a, not available.

**Table 3 nutrients-12-03566-t003:** Cytokine level measured by ELISA test in biopsies treated with the product of in vitro digestion of rice flours (GF) and wheat flours of Saragolla (Sar) or Cappelli (Cap). The values reported are normalized with respect to GF.

	IL-15 Normalized (Mean (Range))	IFN-γ Normalized (Mean (Range))
**Control Group**	*n* = 15	*n* = 14
Sar/GF	0.97 (0.07–3.53)	0.90 (0.48–1.31)
Cap/GF	1.20 (0.12–2.95)	1.29 (0.52–4.97)
**Celiac Group**	*n* = 16	Sar *n* = 22-Cap *n* = 23
Sar/GF	1.48 (0.36–2.51)	1.87 (0.19–17.06)
Cap/GF	1.43 (0.64–2.71)	0.93 (0.12–3.03)

Abbreviations: ELISA, enzyme-linked immunosorbent assay; IL-15, interleukin 15; IFN-γ, interferon γ; n/a, not available.

**Table 4 nutrients-12-03566-t004:** Cytokine level measured by ELISA test in biopsies treated with the product of in vitro digestion of rice flours (GF) and wheat flours of Saragolla (Sar) or Cappelli (Cap). The values reported are normalized with respect to GF. Comparison of duodenal biopsies of one patient before and after three months of the diet with Saragolla pasta.

Timing	IFN Normalized		IL-15 Normalized	
	Sar	Cap	Sar	Cap
before	4.85	n/a	0.55	0.84
after	0.53	3.03	1.05	0.74

Abbreviations: ELISA, enzyme-linked immunosorbent assay; IL-15, interleukin 15; IFN-γ, interferon γ.

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
