# Peer review of "The Diverse Potential of Gluten from Different Durum Wheat Varieties in Triggering Celiac Disease: A Multilevel In Vitro, Ex Vivo and In Vivo Approach"

_nutrients, 2020, doi:10.3390/nu12113566_

Round 1

Reviewer 1 Report

I had the pleasure  to review the original article about different potential of gluten in different durum wheat varieties in triggering celiac disease. It is a well written manuscript focusing on the issue of growing number of celiac disease diagnoses and possible diateary approach to prevention.

Author Response

All the authors are very grateful to the reviewer for the time spent to evaluate the manuscript and for appreciating it.

Reviewer 2 Report

I read with interest the MS. "The Diverse Potential of Gluten from Different
Durum Wheat Varieties in Triggering Celiac Disease: a Multilevel In Vitro, Ex Vivo and In Vivo Approach". Gluten free diet is often poorly tolerated by celiac disease patients and any effort to improve its palatability is welcomed. However I have some major points, namely: a) small sample size do not consent the conslusions' emphasis, I would suggest the Authors to tempre their conclusioons adding the samll sample size among study limitations, b) number of biopsies in the control group is far too little to diagnosis small bowel mucosal abnormalities. I understand concerns about potential bleeding, but this needs to be explained and highlighted in the limitations section, c) To follow-up the patients evaluation of deamidated gliadin antibody IgG class was not performed. Since this is the most sensitive marker for diet compliance it needs to acknowledged and explained.

Minor suggestions: I would avoid quoting the single patient with repeated duodenal biopsy for it makes the reader speculating reasons for not pursuing same policy in the others and do not add to the MS

Author Response

The authors are grateful to the reviewer for the observations and suggestions.

  1. We agree that the small sample cannot drive to strong conclusions, and these results represent the basis for wider studies. Conclusions were tempered as suggested.
  2. We agree that the patients enrolled in the control group underwent a limited number of biopsies (1 in the descending duodenum and 1 in the duodenal bulb), but both histology results and clinical characteristics of these subjects suggest any mucosal alteration of the small intestine. On the contrary, all of the control subjects referred to the Gastroenterology Unit for reasons other than suspected malabsorption. Therefore, the standard number of duodenal biopsies was taken and processed for the histological exam.
  3. For what concerns the follow up, it was set according to both European and American guidelines for celiac disease: “The physician should check the integrity of small-bowel absorption, associated autoimmune conditions (in particular thyroid disorders and T1DM), liver disease and dietary adherence by measuring coeliac-specific antibodies (anti-TG2 or EMA/DGP).” (Al-Toma A, Volta U, Auricchio R, Castillejo G, Sanders DS, Cellier C, Mulder CJ, Lundin KEA. European Society for the Study of Coeliac Disease (ESsCD) guideline for coeliac disease and other gluten-related disorders. United European Gastroenterol J. 2019 Jun;7(5):583-613. doi: 10.1177/2050640619844125. Epub 2019 Apr 13. PMID: 31210940; PMCID: PMC6545713). “Monitoring of adherence to GFD should be based on a combination of history and serology (IgA TTG or IgA (or IgG) DGP antibodies). ” (Rubio-Tapia A, Hill ID, Kelly CP, Calderwood AH, Murray JA; American College of Gastroenterology. ACG clinical guidelines: diagnosis and management of celiac disease. Am J Gastroenterol. 2013 May;108(5):656-76; quiz 677. doi: 10.1038/ajg.2013.79. Epub 2013 Apr 23. PMID: 23609613; PMCID: PMC3706994.)

Finally, we agree that the description of the management of the only patient who underwent duodenal biopsies before and after the experimental diet is isolated and may be misleading, if not correctly interpreted. Anyway, the precise follow up of this patient was included in the manuscript for completeness of the reported data. Moreover, the entire CD-Sar cohort did not undergo a second esophagogastroduodenoscopy due to ethical concerns. As specified in the discussion, “…it would have needed a second invasive exam against any clinical indications that is ethically inappropriate.” Overall, the results obtained with this single patient were considered by the authors important to complete data description, although the concept of the impossibility to draw conclusions on isolated data was reinforced.

Reviewer 3 Report

1. The in-vitro portion of the experiment is quiet complicated; readers would benefit from a concise and clear conlusion at the end of each section.

2. The manuscript is very wordy, especially the Introduction and Methods sections.  On the same note, the manuscript would greatly benefit from english editing with the aims of shortening and phrasing the background and methods more clearly.

3. Throughout the manuscript, the authors go into elaborate explanations of their choices of methods.  These sentences/paragraphs are better suited for the Discussion section.

4. The limitations paragraph in the Discussion should mention lack of placebo-control, the non-random division of patients, and the fact that 50% of patients did not repeat their 3 month serology.

.

Author Response

The authors are grateful to the reviewer for the observations and suggestions.

  1. The in vitro paragraph in the “Materials and Methods section” has been written as concisely as possible, maintaining all the necessary information for completeness. Authors agree that the in vitro paragraph in the “Results” section was quite complicated, maybe due to the extensive description of peptides in the text. The description was left only in Figure 1 legend for completeness.
  2. Thank you for the observation, the authors rephrased the MS as much as possible.
  3. Thank you for the comment, explanations of choices have been moved to the discussion section.

Thank you for your comment, your observations were added to the manuscript.

Round 2

Reviewer 2 Report

I read with interest the revised version of the MS "The diverse potential of gluten from different durum wheat varieties in triggering celiac disease: a multilevel in vitro, ex vivo and in vivo approach" by Gaiani F and coworkers. All of my queries have been addressed in full. No additional suggestions on this side.

This manuscript is a resubmission of an earlier submission. The following is a list of the peer review reports and author responses from that submission.

Round 1

Reviewer 1 Report

This manuscript reports a multilevel approach (in vitro, ex vivo, in vivo) to assess the potential of different durum wheat varieties to trigger celiac disease. It concludes that the variety Saragolla has a limited ability to trigger immune responses. In total, the experimental design is not well thought out and the results do not support the conclusions drawn.

In addition, it is carried out in a much too superficial way in total. It has a bit of a clinical study, it has a bit of an ex vivo model system, it has a bit of proteomics, etc., but none of these parts has been carried out with the scientific depth and rigor needed to be useful.

Although the authors do their best to explain their overall intention, this remains unclear until the very last sentence. In total, the manuscript does tend to suggest that this special variety of durum wheat might, in some obscure way, be tolerable for CD patients. This is nothing else but dangerous.

The main reason why this feeling appears is Figure 3, that, when shown out of context (as often happens), seems to say that the Saragolla diet does not show a significant difference to the GF diet. Again, this is nothing else but dangerous.

Further comments:

Description of experimental setup is not comprehensive and comprehensible:

L31: clarify that the control group are patients with other gastrointestinal problems
L129: clarify criteria for inclusion and exclusion in celiac and control group
L140: clarify that patients have not been on a gluten free diet before the study
L147: It appears that the CD-GFD and CD-SAR were not blinded, because this is probably hard to do. What effect do you think this open study had on the results? This point is not discussed at all.
L150: demonstrate how gluten free diet was verified – especially since newly diagnosed CD patients without prior knowledge on CD cannot be expected to perfectly follow a GFD from day 1. Did they receive nutritional counselling? Did they receive any kind of guidance?
Table 1: add characterization of control group (e.g. tTG IgA and HLA), add total IgA values for assessment

Description of results is not comprehensive and comprehensible:

L106: how were these 82 varieties selected? Based on which criteria?
L201: add characterization of durum wheat varieties (e.g. protein content); add characterization of pasta (e. g. gluten content, peptide content)
Figure 1: add reference of amount values
L208: clarify meaning of figure 2A
L220: clarify criteria for grouping (figure 2B); clarify selection of Saragolla and Cappelli as representative
Figure 2: provide in a better quality
L259: clarify calculation of significances, add significances for DPG-AGA IgG and Hb
Figure 3: clarify which data have been used for this figure, delta tTG values not comprehensible, because taking the values from Table 2 does not add up
Table 3: are there significant differences between Sar and Cap?; check for outliers
L291 explain how you overcame the interferences
SuppTable1: clarify calculation of standard deviation; check positions after decimal point for N

For all tables and figures: Add a detailed descriptions including abbreviations (e.g. table 4 is based on results of one single patient, but this is not clear at all)

The conclusions drawn are not supported by the results, because the results could well also be interpreted in a totally different way
L37/title: clarify “diverse potential”, manuscript contains only results concerning Saragolla
L273: explain how a Marsh grade of 3a could be “good”; or rather how much would you expect a Marsh 3a lesion to worsen? Straight to refractory CD?
L334: check causality (if you are only looking for alpha- and gamma-gliadins, you cannot find anything else)
L367: clarify based on which data you assessed Saragolla not to stimulate immune response

Controversial discussion is missing:
Take into account: time-delayed effect of gluten free diet on damage of mucosa, other diseases of the celiac patients and control patients, other therapies of the patients during study, real number of patients (antibody determination after diet)

The structure of the paper is not clear/information is not cumulated at one point in the paper/information is not clear (e.g. selection of Saragolla and Cappelli -> because of median or something else; were the patients under gluten free diet before the study starts?)

L399: avoid generalization: only results for Saragolla shown

Clarify abbreviations and use them correct and consistently (n/a especially in tables, DPG-AGA IgG, CD-Sar vs. CD-SAR, GFD, TT vs. TP, IL-15)

Check units, spaces between values and units, comma, upper and lower case

Author Response

Dear Reviewer, point-by point answers can be found in the attached file.

Thank you for your suggestions.

Reviewer 2 Report

Summary: The article uses the proteomic analysis of the digestions of the flours of 82 varieties of durum wheat to select two varieties of this species with high and low content of toxic and immunogenic peptides (Senatore Capelli and Saragolla, respectively). These two varieties are used to determine ex vivo the maximum variation in the immunogenic capacity of durum wheat. Likewise, it analyzes the effect of consuming the Saragolla variety in celiac patients.

I think that the study is of great interest, since few works address the study of a sufficient number of varieties with this approach. But I want to draw attention to several aspects of the work:

  1. In my opinion, it should be specified in the title that the varieties studied are durum wheat because, without being untrue, it is misleading. Durum wheat, tetraploid, represents only a small percentage of the wheat grown and consumed in the world (less than 10%), being the bread wheat, hexaploid, the most widely cultivated. Durum wheat lacks the hexaploid wheat D genome which contributes greatly to the amount of immunogenic peptides in wheat. Furthermore, the end use of the two species is also different: bread is normally made with hexaploid wheat and pasta with durum wheat, so both species are not directly comparable. Nevertheless, it should be noted that “durum wheat” is included as a keyword and is mentioned in the abstract.
  2. The number of varieties studied is large enough to analyze the variability for immunotoxicity, but neither a list of the plant material describing at least its origin has been included, nor its value has been justified to represent diversity of that that species in an area or region. That information needs to be included.
  3. In the discussion and conclusions, the term species is used instead of variety (lines 332, 379, 388, 390 and 401). For the above mentioned reasons, both concepts should not be missused.
  4. Supplemental Table 1 does not show the differences between the various peptides, but rather a significant variance due to the variety factor for each peptide. In any case, the differences between the peptides can be seen graphically in Figure 1.
  5. The reference [27] of line 190 should appear in line 195, after (QQPQQPFPQ), since what that paper propose is a list of celiac disease relevant T-cell epitopes recognized by CD4 + T cells, all of them of 9 amino acids.
  6. The PCA chart is illegible. I disagree on the contribution of the different peptides to each dimension, which in each case is the projection of that variable on the corresponding axis. As the relationships between peptides have not been discussed, I am not going to go into that matter further.
  7. Finally, I have not been able to find any data on the protein (total protein or only prolamin) content of the flour samples or of the pasta used in the diet. It is important to know it because the protein content depends not only on the genetic factor (variety) but to a great extent on the environmental effect. In addition to the presence of toxic epitopes defined by the sequence of the different gliadin genes of each sample, the amount of those epitopes differ for a particular variety depending on the environment (including agronomical practices) it is grown. In relation to the study of the different varieties, all have been cultivated under the same conditions, so they are comparable. However, the origin of the plant material used to make the pasta is not clear to me.

Author Response

Dear Reviewer, point-by-point answers can be found in the file attached. 

Thank you for your suggestions.

Reviewer 3 Report

The article aims to assess the potential of gluten from different wheat varieties in triggering CD (in vitro, ex vivo, in vivo). Peptides triggering CD were identified and quantified to discriminate of varieties generating low impact on CD (e.g. Saragolla) and high impact (e.g. Cappelli). They enrolled healthy subjects and celiac patients for the in vivo phase: celiacs were divided into gluten-free diet group (CD-GFD), and GFD added of Saragolla-based pasta group (CD-SAR). Data were compared between CD-SAR and CD-GFD before and after the experimental diet, demonstrating a limited ability of Saragolla to trigger immunity, although not comparable to GFD. Ex vivo studies showed that Saragolla and Cappelli activated immune responses, although with great variability among patients. The authors concluded that Saragolla is not indicated for celiacs, yet it has a limited potential to trigger adverse immune response. Although the relevance of the data is important, some points need to be clarified before considering the manuscript suitable for publication.

General

Results: The immune response of CD can vary according to age, so it can be different in children and adults. For this reason, the article must specify how many children and adults were recruited, and if differences are observed in the results obtained between them. The dynamics of CD antibodies after diagnosis varies according to adherence to the GFD, and to some other factors: type of antibodies, age at diagnosis, coexisting diseases, antibody titers at diagnosis and different assays and techniques available (Sansotta N et al, 2020, Webb C et al, 2015, Bufler P et al, 2015, Benelli E et al, 2016…..).

Discussion: please define the weaknesses and strengths of the study.

Specific Comments:

  • Line 129: “The inclusion criteria were age comprised between 2 and 75 years, ability to sign informed consent (by the legal guardian in case of minors)…” The ability of sign informed consent should not be considered inclusion criteria, but rather an ethical and legal requirement, as the authors’ state in lines 134-138.
  • Line 130 “….elevated tTG IgA (UNL=10U/mL) and or DPG-AGA IgG (UNL=10U/mL)” Please specify the analysis method for tTG and DPG-AGA. What guidelines for diagnoses were used? Why is the DGP-AGA analysis performed and in what cases? Only for children or also in the case of adults? Was EMA done?
  • Figure 2 does not have a good resolution
  • Table 1. Please include a legend with abbreviations (M, F, BMI, CD, n/a…). “Familiarity for CD” First degree???.

“Age”, please include number of children and adults.

“DPG-AGA IgG” How many patients were determined for DGP-AGA? n??? Regarding histology, there are 3 Marsh 1 patients, were they adult or pediatric? If they were pediatric, based on what was the diagnosis made? No Marsh 2 patients?

The authors should include the SD with the mean.

  • Line 241: “At recruitment, tTG IgA levels were elevated and suspicious for CD, with comparable titers between CD-Sar group and CD-GFD group.” Also comparable between children and adults?
  • Table 2 “n/a T0; n/a 3M” It’s not clear, please clarify.
  • Figure 3. “delta tTG” In percentage (%), in x times ULN??? Are units missing or are they relative units?
  • Line 268: “The influence of diet on immune activation was mainly determined by tTG IgA variation, as these antibodies are the gold standard both for diagnosis and follow up, although DPG-AGA IgG and hemoglobin serum levels were investigated, showing similarly no differences between groups.”  IgA TGA work excellently for diagnosis, but these are less accurate for dietary monitoring (Mehta P et al, 2018, Bannister EG et al, 2014).There are other tools to check if compliance with the diet has been correct: GIP and dietary records. Were they used in the study? If they were used, they can be included, if they were not used it, maybe the authors could be put as a weakness of the study.
  • Figure 4 and 5 could join. 
  • Discussion: The first paragraph corresponds to Introduction

Author Response

(The authors gave the same response as above.)

Round 2

Reviewer 1 Report

The authors have done their best to address the changes requested and the manuscript is improved a lot compared to the first version. However, due to the experimental design, my main point of criticism still holds true that it is carried out in a much too superficial way in total. The authors’ explanation on trying to combine results and insights from different models makes sense in a way, but looking at everything without looking at some experiments in greater detail may lead to erroneous conclusions. The clarification that the authors tried to look for less immunoreactive durum wheat varieties for the general population to sort of prevent people from developing celiac disease is a hypothesis that has not been proven yet. It is not known whether this is at all relevant to the healthy population and whether this could really help prevent the onset of celiac disease. From all studies so far, it does not seem likely that this prevention scenario works at all.

Reviewer 2 Report

After reading the revised article, I have verified that my comments have been taken into account. I have no further observations to make.